# Aerobic Exercise Attenuates Doxorubicin-Induced Cardiomyopathy by Suppressing NLRP3 Inflammasome Activation in a Rat Model

**DOI:** 10.3390/ijms25179692

**Published:** 2024-09-07

**Authors:** Phichaya Suthivanich, Worakan Boonhoh, Natticha Sumneang, Chuchard Punsawad, Zhaokang Cheng, Sukanya Phungphong

**Affiliations:** 1Doctor of Philosophy Program in Physiology, Graduate School, Chulalongkorn University, Bangkok 10330, Thailand; phichaya.sut@gmail.com; 2Akkhraratchakumari Veterinary College, Walailak University, Nakhon Si Thammarat 80161, Thailand; worakan.bo@mail.wu.ac.th; 3Department of Medical Sciences, School of Medicine, Walailak University, Nakhon Si Thammarat 80160, Thailand; natticha.su@wu.ac.th (N.S.); chuchard.pu@wu.ac.th (C.P.); 4Research Center in Tropical Pathobiology, Walailak University, Nakhon Si Thammarat 80160, Thailand; 5Department of Pharmaceutical Sciences, Washington State University, Spokane, WA 99202, USA; zhaokang.cheng@wsu.edu

**Keywords:** exercise, cardiomyopathy, doxorubicin, NLRP3 inflammasome, inflammation

## Abstract

Doxorubicin (DOX) is a potent chemotherapeutic agent with well-documented dose-dependent cardiotoxicity. Regular exercise is recognized for its cardioprotective effects against DOX-induced cardiac inflammation, although the precise mechanisms remain incompletely understood. The activation of inflammasomes has been implicated in the pathogenesis and treatment of DOX-induced cardiotoxicity, with the nucleotide-binding domain-like receptor protein 3 (NLRP3) inflammasome emerging as a key mediator in cardiovascular inflammation. This study aimed to investigate the role of exercise in modulating the NLRP3 inflammasome to protect against DOX-induced cardiac inflammation. Male Sprague–Dawley rats were randomly assigned to receive a 10-day course of DOX or saline injections, with or without a preceding 10-week treadmill running regimen. Cardiovascular function and histological changes were subsequently evaluated. DOX-induced cardiotoxicity was characterized by cardiac atrophy, systolic dysfunction, and hypotension, alongside activation of the NLRP3 inflammasome. Our findings revealed that regular exercise preserved cardiac mass and hypertrophic indices and prevented DOX-induced cardiac dysfunction, although it did not fully preserve blood pressure. These results underscore the significant cardioprotective effects of exercise against DOX-induced cardiotoxicity. While regular exercise did not entirely prevent DOX-induced hypotension, our findings demonstrate that it confers protection against DOX-induced cardiotoxicity by suppressing NLRP3 inflammasome activation in the heart, underscoring its anti-inflammatory role. Further research should explore the temporal dynamics and interactions among exercise, pyroptosis, and other pathways in DOX-induced cardiotoxicity to enhance translational applications in cardiovascular medicine.

## 1. Introduction

Cardiovascular disease remains the leading cause of morbidity and mortality globally, necessitating continued research into the mechanisms underlying cardiovascular function [1]. While the cardiovascular benefits of regular exercise, such as enhanced cardiac function and reduced cardiovascular risk, are well documented [2,3,4], the precise molecular pathways through which exercise exerts its cardioprotective effects, particularly against cardiotoxic agents, are still under investigation. Doxorubicin (DOX), a widely used chemotherapeutic, is limited by its dose-dependent cardiotoxicity, characterized by oxidative stress, inflammation, and myocardial dysfunction [5,6,7,8]. Emerging evidence suggests that exercise may modulate these harmful pathways, potentially offering protection against DOX-induced cardiotoxicity.

Pyroptosis is an inflammatory form of programmed cell death dependent on caspases −1, −3, −4, −5, and −11 and is closely linked to inflammasome activation [9]. Among inflammasomes, the nucleotide-binding oligomerization domain-like receptor protein 3 (NLRP3) inflammasome plays a pivotal role in mediating inflammation and cellular damage in cardiovascular diseases. This multiprotein complex responds to cellular stress signals by activating pro-inflammatory cytokines, such as interleukin-1β (IL-1β) and interleukin-18. Dysregulation of NLRP3 inflammasome activity has been implicated in various cardiovascular conditions, including heart failure, atherosclerosis, and myocardial infarction [10,11,12]. Given these associations, our study aimed to explore the interactions between exercise, DOX-induced cardiotoxicity, and the NLRP3 inflammasome. We specifically examined the effects of treadmill running on cardiovascular function in a rat model of DOX-induced cardiotoxicity. This work extends previous research on the cardiovascular benefits of exercise to the specific context of NLRP3 inflammasome-regulated chemotherapy-induced cardiotoxicity.

Previous studies have highlighted the protective effects of exercise against cardiovascular diseases, particularly through anti-inflammatory and antioxidant pathways [13,14,15]. However, the role of the NLRP3 inflammasome in mediating the cardioprotective effects of exercise, especially under DOX-induced stress, remains underexplored. We hypothesize that exercise intervention can attenuate DOX-induced cardiotoxicity by modulating NLRP3 inflammasome activity. Investigating these pathways may offer new insights into the mechanisms of exercise-induced cardiovascular protection and could inform the development of targeted therapeutic strategies to manage DOX-induced cardiotoxicity, ultimately improving patient outcomes.

## 2. Results

### 2.1. General Characteristics and Cardiac Morphology Associated with Dox Treatment

Thirty-two male Sprague–Dawley rats were randomly allocated to either a sedentary or an exercise group. Following a 10-week regimen of moderate-intensity exercise or sedentary activity, all animals received saline or doxorubicin hydrochloride (1 mg/kg body weight/day) for ten consecutive days. Blood pressure was measured one week before and one week after the administration of saline or doxorubicin. Cardiac structure and contractile function were assessed ten days following the final doxorubicin injection. Figure 1 provides a detailed overview of the experimental methodology.

Body weight, heart weight, kidney weight, soleus weight, and tibial length were comparable across the sedentary and exercised rats, with and without doxorubicin administration (Table 1). Moderate-intensity treadmill running induced physiological cardiac hypertrophy, as indicated by increases in heart mass, the heart weight-to-body weight ratio, and the heart weight-to-tibial length ratio. Doxorubicin treatment led to a decrease in body weight, heart weight, and kidney weight, without significantly affecting tibial length. The heart weight-to-tibial length ratio, a marker of cardiac hypertrophy, was significantly increased in the saline-exercise group, whereas it was reduced in the doxorubicin-sedentary group. Notably, regular exercise prevented the DOX-induced decrease in the hypertrophic index (HW/TL ratio, Table 1).

A similar protective effect of exercise was evident at the microscopic level. The cross-sectional area and minimal Feret’s diameter of cardiomyocytes in the ventricles were significantly greater in the exercise-trained group compared to the sedentary group. Regular exercise notably mitigated the cardiomyocyte size alterations induced by DOX (Figure 2A–C). Furthermore, heart tissue sections from the DOX-treated rats exhibited cytoplasmic vacuolization and myocytolysis (disruption of myofibrils), which were effectively prevented by regular exercise (Figure 2D).

### 2.2. Cardiac Systolic Function and Blood Pressure Properties of Doxorubicin-Treated Rats

Cardiovascular function was assessed using echocardiography and non-invasive blood pressure measurements (Figure 3 and Figure 4). Neither exercise training nor DOX injection alone significantly affected the posterior wall or interventricular septal thickness, nor were there notable differences in left ventricular (LV) mass or the LV mass-to-tibial length ratio between groups (Table 2). As anticipated, DOX treatment led to systolic dysfunction, evidenced by an increased LV internal diameter at the end-systole, and reductions in ejection fraction and fractional shortening (Table 2, Figure 3B,C). Regular exercise mitigated the systolic dysfunction induced by DOX. Additionally, rats treated with DOX exhibited significantly lower systolic, diastolic, and mean arterial blood pressures compared to the saline group (Figure 4A–C). The reduced blood pressure in DOX-treated rats, coupled with a lack of heart rate acceleration, suggested an impaired autonomic response (Figure 4D). Collectively, these findings indicate that regular exercise effectively counteracts the decline in arterial blood pressure associated with DOX-induced cardiotoxicity.

### 2.3. NLRP3 Signaling Activation in the Heart after DOX Treatment

Given that DOX can induce adverse effects through inflammation and cardiac apoptosis, we aimed to evaluate the impact of exercise on pyroptosis-related cardiotoxicity induced by DOX treatment. To assess this, we measured the expression levels of pyroptosis-associated proteins, including NLRP3, cleaved caspase-1, and cleaved IL-1β in ventricular tissue (Figure 5). Our results indicated that DOX treatment significantly elevated the protein levels of NLRP3, cleaved caspase-1, and cleaved IL-1β compared to the saline-sedentary group (Figure 5B–D). Although regular exercise did not directly alter the expression levels of these proteins associated with NLRP3 inflammasome activation, it partially mitigated the increase in NLRP3, cleaved caspase-1, and cleaved IL-1β induced by DOX (Figure 5B–D). These findings suggest that exercise has a suppressive effect on DOX-induced NLRP3 activation.

We subsequently investigated whether changes in NLRP3 expression could serve as a marker for cardiomyopathy and examined its relationship with cardiac function (Figure 6). Our analysis revealed that NLRP3 expression was strong inversely correlated with cardiac ejection fraction (R^2^ = 0.5052, *p* = <0.0001) (Figure 6) but weak inversely correlated with blood pressure (R^2^ = 0.1417, *p* = 0.0442) (Appendix A). Conversely, NLRP3 expression showed a direct weak correlation with histopathological score (R^2^ = 0.2342, *p* = 0.0105) (Appendix A).

## 3. Discussion

Our findings highlight the potential of regular exercise in preventing the upregulation of NLRP3 inflammasome activation during DOX-induced cardiotoxicity. We observed that the level of NLRP3 expression was closely associated with cardiac morphological changes and contractile dysfunction. The suppression of NLRP3 inflammasome by regular exercise appears to play a crucial cardioprotective role, offering significant protection against DOX-induced cardiotoxicity.

Doxorubicin induces oxidative stress, inflammation, and apoptosis [16,17,18,19,20,21], which may have complex interactions with pyroptosis, extending beyond the mechanisms modulated by exercise. As a potent chemotherapeutic agent, DOX activates the NLRP3 inflammasome signaling cascade associated with pyroptosis. Dysregulation of this pathway has been implicated in a range of cardiovascular disorders, including DOX-induced cardiotoxicity [6,8,18,19,21]. Moreover, survivors of anticancer treatments face elevated cardiovascular risks, including ischemic heart disease, heart failure, and endothelial dysfunction, which are often exacerbated by chronic inflammation and metabolic disorders induced by these treatments. This underscores the importance of cardiovascular monitoring and the potential need for tailored interventions in this population [22]. Our current data demonstrate that the DOX-treated rats exhibited not only cardiac atrophy, systolic dysfunction, and hypotension but also impaired autonomic control (Figure 2, Figure 3 and Figure 4). Afonso et al. (2023) suggested that the cardiotoxic effects of DOX may involve chemoreceptor dysfunction [23]. Moreover, Shi et al. (2023) reported that DOX-induced oxidative stress and inflammation could disrupt brain signaling along the autonomic circuits, leading to impaired autonomic regulation [17]. In DOX-induced cardiotoxicity, the reduced chemoreflex response may explain the observed hypotension and unchanged heart rate in the DOX-treated group. These hypotensive effects could be attributed to direct cardiac toxicity, as DOX is known to impair cardiac muscle function and disrupt calcium homeostasis, leading to reduced cardiac inotropism and compromised heart pumping efficiency [24]. This diminished myocardial contractility likely reduces vascular resistance, lowering blood pressure. Additionally, DOX has been linked to endothelial dysfunction, characterized by impaired vasodilation and increased vasoconstriction, further affecting blood pressure regulation [25]. These findings underscore the multifaceted impact of DOX on cardiovascular health and highlight the importance of exploring strategies to mitigate its adverse effects.

Exercise is a widely recognized non-pharmacological intervention that plays a critical role in cardioprotection, primarily through its anti-inflammatory and antioxidant effects. The intricate relationship between aerobic exercise and pyroptosis in various cell types, including the endothelial cells, adipocytes, and hippocampal cells, is increasingly understood as a key factor in the pathogenesis and management of several diseases [26,27,28,29]. Aerobic exercise has protective effects on endothelial cells, especially in cardiovascular diseases. Endothelial dysfunction, linked to both traditional and emerging risk factors like aging and hypertension, is an early marker of cardiovascular issues [30]. Our study may support the view that lifestyle interventions, particularly exercise, are crucial for restoring endothelial function by reducing inflammation through the inhibition of NLRP3 inflammasome activation. For instance, Lee et al. (2018) found that voluntary wheel running improved NLRP3 inflammasome activation and IL-1β processing in coronary arterioles of high-fat-diet-induced obese rats by restoring nitric oxide signaling and reducing oxidative stress [26]. This finding underscores the potential of aerobic exercise to mitigate endothelial dysfunction by targeting key inflammatory pathways. Similarly, in adipocytes, where pyroptosis is closely associated with metabolic disorders such as obesity and diabetes, aerobic exercise has been shown to reduce the expression of pyroptosis-related factors. These effects highlight the broader metabolic benefits of exercise beyond weight management, extending to the cellular mechanisms that drive inflammation and insulin resistance [27,28,31].

In addition to its effects on pyroptosis, aerobic exercise has been shown to influence autonomic regulation. Unlike strength training, aerobic exercise has been reported to reduce resting heart rate and enhance high-frequency respiratory-related variability, which may contribute to its cardioprotective effects by improving autonomic function [32]. This aligns with our findings that aerobic exercise mitigated DOX-induced hypotension and helped to maintain blood pressure in the treated rats (Figure 4). Collectively, the evidence suggests that aerobic exercise is a versatile and effective non-pharmacological approach that exerts protective effects against pyroptosis across diverse cell types. This positions aerobic exercise as a key strategy in the prevention and management of diseases associated with pyroptosis, including cardiovascular and metabolic disorders, as well as neurodegenerative conditions.

The anti-inflammatory effects of exercise on DOX-induced NLRP3 inflammasome activation are multifaceted and not entirely elucidated. Our study aligns with prior research [26], indicating that exercise does not modulate NLRP3 inflammasome activation under normal physiological conditions. Despite inducing physiological hypertrophy in the rat heart, the levels of key inflammasome markers such as NLRP3, cleaved caspase-1, and cleaved IL-1β remained unchanged, suggesting that hypertrophic responses to exercise occur independently of the NLRP3 inflammasome pathway. Despite DOX-induced stress, this lack of change in the NLRP3 inflammasome activation markers in response to treadmill exercise underscores the broad cardiopulmonary benefits of aerobic exercise beyond those provided by weight training. These findings suggest that aerobic exercise enhances cardiovascular function through mechanisms that do not involve the NLRP3 inflammasome under non-stressful conditions. Our findings revealed that DOX-treated rats subjected to exercise exhibited lower levels of pyroptosis-associated proteins than their sedentary counterparts. Furthermore, while exercise was able to mitigate the DOX-induced decline in systolic function, it did not fully restore it to normal. One plausible explanation is that while exercise effectively suppresses the expression of pyroptosis-associated proteins, this suppression may not be sufficient to counteract the broader array of pathological mechanisms triggered by DOX. DOX-induced cardiotoxicity involves oxidative stress, apoptosis, and mitochondrial dysfunction, which may surpass the protective effects of exercise on the inflammasome pathway. While exercise reduces certain aspects of DOX-induced inflammation, these complex mechanisms may limit its overall effectiveness. Further research is needed to elucidate how exercise interacts with these pathways and to develop comprehensive strategies combining exercise with other interventions to better mitigate DOX-related cardiac damage.

The results of the present study demonstrated an inverse correlation between NLRP3 expression and cardiovascular function, suggesting that the cardioprotective effects of exercise may regulate NLRP3 inflammasome activation in the heart. We propose several potential inhibitory pathways. First, exercise may exert anti-inflammatory effects by stimulating anti-inflammatory cytokines, such as interleukin-10 (IL-10), and transforming growth factor-beta (TGF-β) [33]. These cytokines inhibit inflammasome activation and may counteract the pro-inflammatory signals DOX triggers. Furthermore, DOX-induced cardiotoxicity is associated with increased oxidative stress, a known trigger for inflammasome activation. Since redox signaling is a primary driver of NLRP3 inflammasome activation, enhancing endogenous antioxidant defenses through exercise may reduce oxidative stress and suppress NLRP3 inflammasome activation [34]. Second, exercise has been implicated in the modulation of autophagy, a crucial cellular process for maintaining homeostasis. Enhanced autophagic activity may facilitate the removal of damaged cellular components, thereby mitigating the accumulation of inflammasome-activating signals and suppressing NLRP3 inflammasome activation [35]. Additionally, mitochondrial dysfunction is known to play a role in NLRP3 inflammasome activation. The improvements in mitochondrial function induced by treadmill running may reduce the release of mitochondrial damage-associated molecular patterns (DAMPs) thus attenuating inflammasome activation. Exercise also activates AMP-activated protein kinase (AMPK), a key cellular energy sensor, which may influence NLRP3 inflammasome activity by regulating cellular energy status and inhibiting pro-inflammatory signaling cascades [36]. Zhang et al. (2023) reported that exercise training attenuates isoproterenol-induced cardiac inflammation by inhibiting the reactive oxygen species (ROS)–NLRP3 inflammasome pathway in an AMPK-dependent manner [36]. Finally, treadmill running may induce epigenetic changes, such as DNA methylation and histone acetylation, which could influence gene expression during inflammasome activation and contribute to the suppression of NLRP3 inflammasome activation [37]. Although these factors were not measured in the current study, they represent attractive targets for future research. For instance, further investigation is needed to confirm the effects of exercise on NLRP3 inflammasome pathways in cardiomyocytes, which can sense the chemical and mechanical alterations associated with DOX-induced cardiomyopathy.

While the beneficial effects of regular exercise in preventing cardiotoxicity have been widely reported [38], our findings indicate that treadmill running in DOX-treated rats did not fully reverse the decline in cardiac systolic function. Therefore, the duration, intensity, and mode of exercise may influence the extent of pyroptosis suppression and thus optimizing these exercise parameters may be crucial for enhancing cardioprotective effects. Moreover, the cardiotoxic effects of DOX involve complex interactions within the cardiac microenvironment, potentially implicating other pathways or cellular processes that are not directly influenced by changes in pyroptosis-associated proteins. The interplay between inflammation, apoptosis, and pyroptosis in response to DOX-induced stress may involve compensatory mechanisms or downstream signaling events that mitigate the impact of exercise-induced pyroptosis suppression.

Considering these complexities, future studies should explore the crosstalk between exercise, pyroptosis, and other cellular pathways involved in DOX-induced cardiotoxicity. Additionally, a comprehensive understanding of the temporal dynamics of exercise-induced molecular adaptations and their implications for long-term cardioprotection is essential. The findings of this study open avenues for further research to decipher the intricate molecular landscape underlying the interplay between exercise and DOX-induced cardiotoxicity, offering valuable insights for developing targeted interventions in cardiovascular medicine.

It is important to acknowledge several limitations of this study. First, our research primarily focused on the NLRP3 inflammasome within cardiac tissue; however, anthracyclines like DOX are known to induce cell death and damage beyond cardiomyocytes, affecting non-myocyte cells (e.g., endothelial cells) and other organs. This widespread cellular damage can initiate compensatory mechanisms not fully explored in this study. Additionally, the systemic effects of exercise on inflammation were not investigated, which could provide valuable insights into the broader implications of exercise in mitigating DOX-induced cardiotoxicity.

Another critical aspect is the exclusive use of low-intensity, moderate treadmill running in the study. While this form of exercise has its merits, the cardioprotective effects of exercise might be less effective at shielding the heart from DOX-induced damage at lower intensities. Previous studies have shown that aerobic exercise, particularly when applied before, during, or after DOX administration, can attenuate cardiotoxicity by improving systolic and diastolic functions [39,40,41,42]. Even acute exercise has been reported to improve survival rates in mice subjected to DOX treatment [43]. Acute exercise protects against doxorubicin-induced cardiac dysfunction by modulating non-enzymatic antioxidants [44]. It increases the expression of redox effector factor-1 (Ref1) and nuclear factor erythroid 2-related factor 2 (Nrf2), enhancing mitochondrial H₂O₂ production and antioxidant enzyme activities. This exercise-induced oxidative stress activates Nrf2, which reduces ROS production and boosts antioxidant defenses [45]. Elevated levels of NADPH oxidase-4 further activate Nrf2, improving the nuclear transcription of antioxidant genes [46]. These adaptations help maintain redox balance in cardiomyocytes and reduce their susceptibility to doxorubicin-induced damage. Further research is needed to explore the long-term sustainability of exercise-induced benefits and the effects of intermittent or discontinued exercise on cardiac NLRP3 inflammasome activity. Despite these benefits, the present study found that while exercise did not entirely prevent DOX-induced cardiac systolic dysfunction or directly modulate NLRP3 inflammasome activation under physiological conditions, it did attenuate the DOX-induced increases in NLRP3, cleaved caspase-1, and cleaved IL-1β expression. These findings suggest that exercise exerts its cardioprotective effects, at least in part, by suppressing the pyroptosis-associated proteins. This suppression is likely mediated through multiple mechanisms, including anti-inflammatory effects, antioxidant activity, autophagy modulation, and mitochondrial function improvements. However, the specific pathways through which these protective effects are exerted remain to be fully elucidated. Considering these limitations, this study underscores the need for further research to explore the complex interactions between exercise, pyroptosis, and other cellular pathways involved in DOX-induced cardiotoxicity. Such research is essential for refining and optimizing exercise protocols to enhance their protective effects and developing targeted therapeutic strategies to mitigate the adverse cardiovascular outcomes associated with DOX treatment.

## 4. Materials and Methods

### 4.1. Ethical Approval

All animal experiments were conducted in accordance with the Guide for the Care and Use of Laboratory Animals (8th edition, NIH) and were approved by the Animal Ethics Committee of Walailak University (WU-ACUC-65062). In addition, this study followed the ARRIVE standards. All approaches adhered to the appropriate standards and regulations.

### 4.2. Materials

All chemicals were purchased from Loba Chemie (Mumbai, India), HiMedia (Mumbai, India), and Sigma-Aldrich (St. Louis, MO, USA). Electrophoretic reagents were purchased from Bio-Rad (Hercules, CA, USA) and Thermo Fisher Scientific (Waltham, MA, USA). Doxorubicin hydrochloride was purchased from the Maharaj Nakhon Si Thammarat Hospital (Pacific Healthcare, Bangkok, Thailand).

### 4.3. Animal

This study exclusively utilized male rats, based on evidence suggesting that female sex hormones may enhance the effects of exercise, as demonstrated by Phungphong et al. (2020) [24]. While acknowledging the importance of gender inclusion in scientific research, we aimed to minimize confounding variables and maintain experimental integrity by focusing on a single sex. This deliberate choice enables a clearer understanding of the specific mechanisms underlying exercise responses.

Male Sprague–Dawley rats, aged 8 weeks and weighing between 250 and 280 g (n = 32), were procured from Nomura Siam International Co., Ltd., Bangkok, Thailand. The animals were housed in groups of four or five per cage under controlled conditions (22 ± 3 °C, 50–60% humidity, and a 12:12 h light–dark cycle) and were provided food and water ad libitum. After a one-week acclimatization period, the rats were randomly assigned to one of the following four groups: (1) a control group receiving saline injections and maintained in a sedentary state (Saline-Sed, n = 8); (2) an exercise group receiving saline injections and undergoing moderate-intensity exercise training (Saline-Ex, n = 8); (3) a group with doxorubicin-induced cardiomyopathy maintained in a sedentary state (DOX-Sed, n = 8); and (4) a group with doxorubicin-induced cardiomyopathy undergoing exercise training (DOX-Ex, n = 8).

### 4.4. Exercise Training Program

After one week of acclimatization, the rats were randomly assigned to the sedentary or exercise training groups. The exercise protocol involved a nine-week running regimen on a motor-driven treadmill, starting with 10 min of running at 21 m/min with a 0% grade on the first day at twilight. The running duration progressively increased to 15, 20, and 25 min on subsequent days. From the second to the tenth week, the rats ran for 30 min at 21 m/min with a 5% grade for five days a week, with an exercise intensity of 65–75% of the maximum oxygen consumption [47]. The running environment included dim light, controlled temperature (25 °C), and a cool air blower for stimulation instead of an electrical shock, as previously described [24,48,49]. The experimental procedures were performed 48 h after the last training session to avoid any acute effects of the last exercise session.

Following the final exercise session, both DOX-treated groups received intraperitoneal injections of doxorubicin hydrochloride (1 mg/kg body weight/day) for ten consecutive days (total dose: 10 mg/kg BW) [50]. The control group received a 0.9% saline injection. During the DOX injections, the rats underwent low-intensity running (0% grade, speed 21 m/min, 10 min twice/day) for an additional 10 days, five days per week.

### 4.5. Non-Invasive Blood Pressure Measurement

Blood pressure and heart rate were measured before and after DOX injection using a tail-cuff sphygmomanometer (CODA monitor, Kent Scientific Corporation, Torrington, CT, USA). Rats were placed in plastic restrainers with a controlled temperature of 33–35 °C using a heating pad. After a 5 min warming period, the blood pressure was recorded by inflating the occlusion cuff to 250 mmHg and deflating it over 20 s. The VPR sensor cuff detected tail volume changes as blood returned during cuff deflation, with a minimum volume change set at 15 μL. The recording sessions included 15–25 cycles per set, excluding the first five acclimation cycles. Rats were habituated for at least five consecutive days before the baseline blood pressure measurements [51].

### 4.6. Echocardiographic Measurement

The rats were anesthetized with sodium pentobarbital (60 mg/kg, IP), and the anesthetized condition was confirmed by checking the toe reflex. Rats were placed on a temperature-controlled plate, their chests were shaved and then cleaned with 70% alcohol. Two-dimensional (2D)-guided M-mode echocardiography was used to measure cardiac geometry and function. The procedure was completed within 30 min and measured the left ventricular (LV) wall thickness, interventricular septum (IVS), posterior wall (PW), and internal dimensions during diastole and systole (LVIDd and LVIDs). LV contractility parameters, including fractional shortening, ejection fraction, and LV mass, were calculated using M-mode measurements [13].

### 4.7. Histopathological Examination

Following echocardiographic examination, the rats were euthanized by inducing deep anesthesia with an anesthetic overdose, followed by cervical capitation to ensure humane euthanasia and, subsequently, the heart was rapidly excised. The hearts were then washed with phosphate-buffered saline (PBS) solution, fixed in 10% formalin, and embedded in paraffin. Next, 5 µm thick sections were prepared using a microtome and placed on microscope slides. Changes in cardiomyocytes were evaluated using histological myocardial sections stained with hematoxylin and eosin (H&E). Cardiomyocyte cross-sectional area (CSA) and minimal Feret diameter were calculated using a digital microscope with ImageJ version 1.53t (National Institutes of Health, Bethesda, MD, USA). Only cardiomyocytes with nuclei, obvious cell borders, and round or rectangular morphology (length:width ratio = 1.5) were examined. The cardiomyocyte CSA was calculated from 15 cells per imaging field, using ten random fields for each heart (i.e., 150 cells per rat heart). A scoring method was used to assess cytoplasmic vacuolization and myofibrillar loss [52]. The histopathological extent was scored on a scale of 0–3, where (0) indicated no signs of myofibrillar loss and cytoplasmic vacuolization; (1) <5% cells had early myofibrillar loss or cytosolic vacuolization; (2) 5–30% cells had marked myofibrillar loss and/or cytoplasmic vacuolization; and (3) >30% had diffuse cell damage, with most cells showing a marked loss of contractile elements and myofibrillar disruption. A histopathologist conducted a blinded evaluation of the slides.

### 4.8. Western Blot Analysis

Frozen ventricular muscle was mixed and homogenized with a RIPA buffer (50.36 mM HEPES, 150.58 mM NaCl, 1.98 mM EDTA, 12.06 mM Sodium Deoxycholate, 1% NP-40 (IGEPAL), pH 7.2), supplemented with 0.5% Triton X-100 and protease and phosphatase inhibitors (Thermo Fisher Scientific, Waltham, MA, USA). The homogenate was separated by sodium dodecyl sulfate-polyacrylamide gel electrophoresis (SDS-PAGE) in 12–15% polyacrylamide gels, transferred onto Hydrophobic Polyvinylidene difluoride (PVDF) membranes, which were incubated in a blocking buffer for 2 h at room temperature, and then incubated overnight at 4 °C with primary antibodies such as rabbit anti-NLRP3 (AB263899, Abcam, 1:1000), rabbit anti-pro-caspase-1 with p10 and p12 (AB179515, Abcam, 1:1000), and rabbit anti-IL-1β (AB254360, Abcam, 1:1000). Goat anti-rabbit immunoglobulin G (IgG) was used at 1:10,000 (AB97051, Abcam, Cambridge, UK). Glyceraldehyde-3-phosphate dehydrogenase (AB181603, Abcam, 1:7500) was used as a loading control to normalize the results. Band density was analyzed using ImageJ version 1.53t (National Institutes of Health, Bethesda, MD, USA).

### 4.9. Statistical Analysis

Investigators were blinded to the group allocation during the endpoint analyses. GraphPad Prism 9 (GraphPad Software Inc., San Diego, CA, USA) was used for statistical analysis and the results are expressed as mean ± standard error of the mean (SEM). The sample size was estimated based on previously published data [53] using the MINITAB Statistical Analysis Package. Normality was assessed using the Shapiro–Wilk test. Differences among multiple groups were analyzed using one-way analysis of variance (ANOVA) with Tukey’s test. *p* values < 0.05 are considered statistically significant.

## 5. Conclusions

In conclusion, our findings contribute to the evolving narrative on the multifaceted benefits of exercise in maintaining cardiovascular health during chemotherapy. Delineation of exercise-mediated mechanisms offers a foundation for targeted therapeutic strategies to mitigate DOX-induced cardiotoxicity and provides valuable insights into translational applications in cardiovascular medicine. Further research is required to elucidate the specific pathways and mediators that orchestrate the protective effects of exercise in the landscape of DOX-induced cardiovascular compromise.

## Figures and Tables

**Figure 1 ijms-25-09692-f001:**
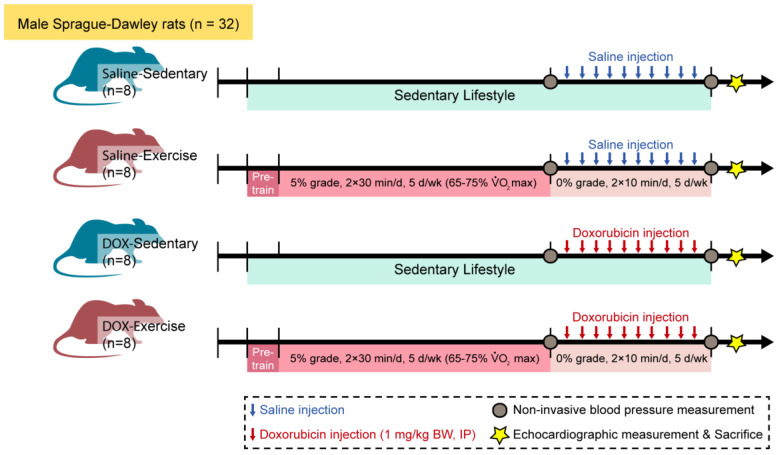
Schematic of the experimental timeline of injection, training, and measurement for animal studies. Male Sprague–Dawley rats were divided into four groups: saline-sedentary, saline-exercise, doxorubicin-sedentary, and doxorubicin-exercise. BW, body weight; d, day; IP, intraperitoneal; VO_2_ max, maximal oxygen consumption. Ten days after the last DOX injection, rats were anesthetized to measure echocardiographic parameters and then euthanized.

**Figure 2 ijms-25-09692-f002:**
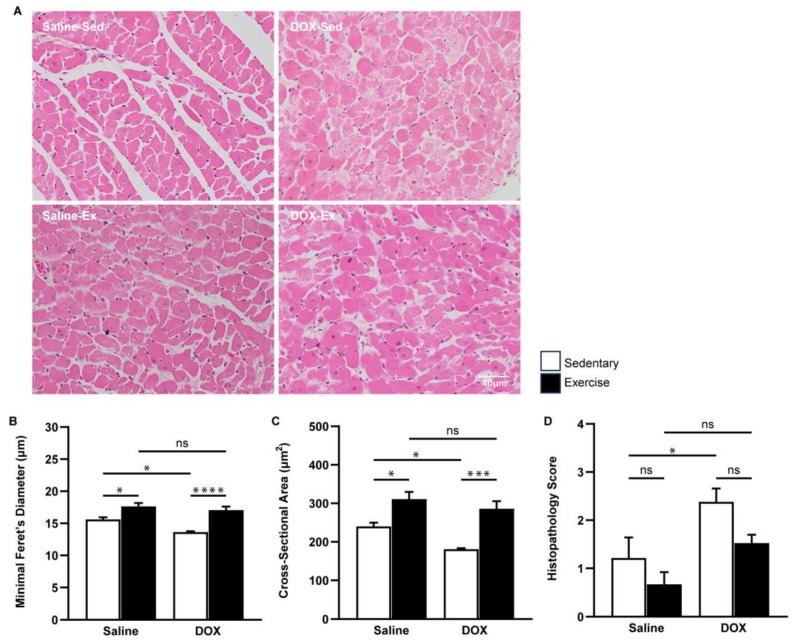
The effect of regular exercise on the cardiac morphology of doxorubicin-induced cardiomyopathy in male Sprague–Dawley rats. Representative hematoxylin and eosin-stained images of the heart sections are shown at 400X (scale bar = 100 μm) (**A**). Minimal Feret’s diameter (fiber diameter) (**B**) and cross-section area (fiber area) (**C**) of cardiomyocytes were calculated using measurements from 150 cardiomyocytes across eight rats per group. Semi-quantitative vacuolization and myofibrillar degeneration of the cardiomyocytes were evaluated by scoring scales from six fields per rat and eight rats per group (**D**). The data are presented as mean ± SEM for eight rats per group. * *p*-value < 0.05, *** *p*-value < 0.001, **** *p*-value < 0.0001 using one-way analysis of variance (ANOVA) followed by Tukey’s test. ns, not significant.

**Figure 3 ijms-25-09692-f003:**
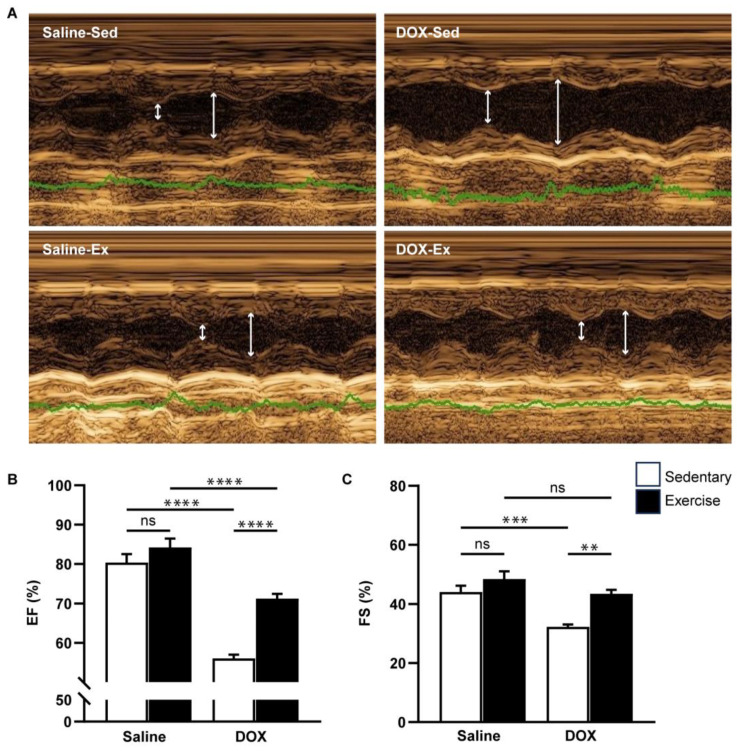
The effect of regular exercise on cardiac systolic function of doxorubicin-induced cardiomyopathy in male Sprague–Dawley rats. Representative images were obtained using M-mode echocardiography (**A**). The ejection fraction (**B**) and fractional shortening (**C**) percentages were averaged over three consecutive cardiac cycles. The data are presented as mean ± SEM for six–eight rats per group. ** *p*-value < 0.01, *** *p*-value < 0.001, **** *p*-value < 0.0001 using one-way analysis of variance (ANOVA) and Tukey’s test. White arrows indicate left ventricular diameter during systole and diastole, respectively. Abbreviated terms include IVSd, interventricular septum during diastole; LVPWd, left ventricular posterior wall during diastole; LVIDs, left ventricular internal diameter during systole; LVIDd, left ventricular internal diameter during diastole; LV Mass, left ventricular mass; TL, tibial length; RWT, relative wall thickness; FS, functional shortening; EF, ejection fraction. ns, not significant.

**Figure 4 ijms-25-09692-f004:**
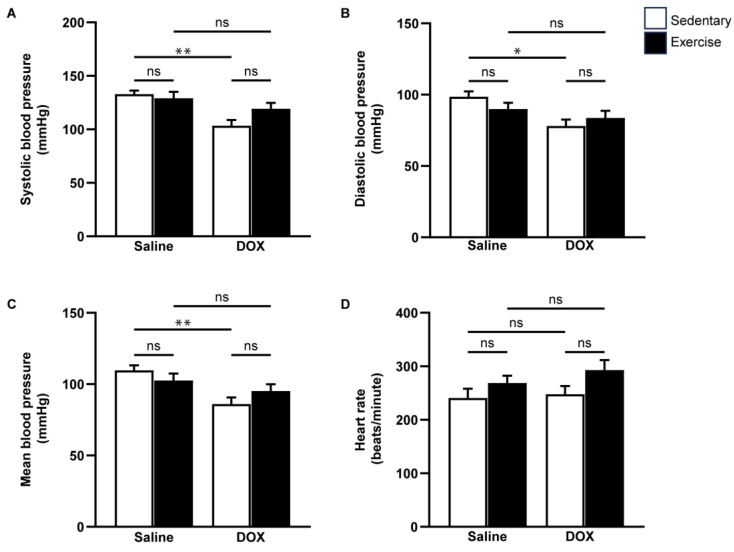
The effect of regular exercise on blood pressure responses after doxorubicin-induced cardiomyopathy in male Sprague–Dawley rats. Systolic blood pressure (**A**), diastolic blood pressure (**B**), mean arterial blood pressure (**C**), and heart rate (**D**) were measured from the tail of each rat using a CODA non-invasive blood pressure device. The data are presented as mean ± SEM for eight rats per group. * *p*-value < 0.05, ** *p*-value < 0.01 using one-way analysis of variance (ANOVA) and Turkey’s test. ns, not significant.

**Figure 5 ijms-25-09692-f005:**
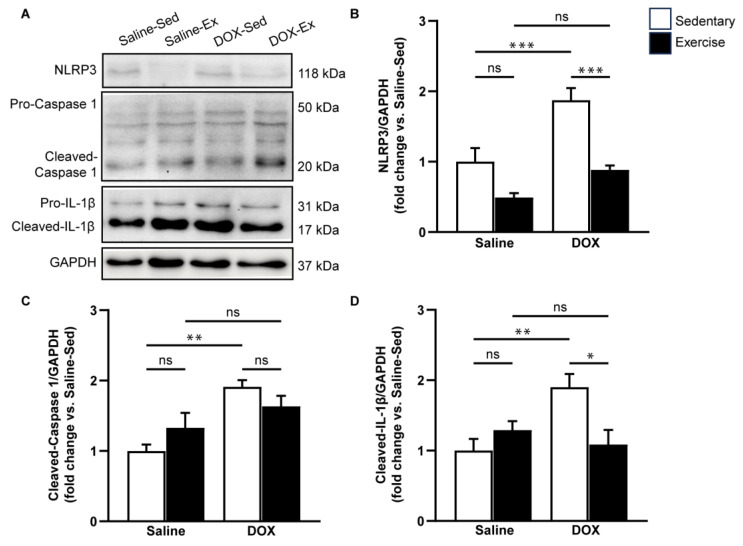
The effect of doxorubicin and exercise training on the expression of the NLRP3 inflammasome signaling pathway. Representative immunoblots of NLRP3, caspase-1, and IL-1β in comparison with GAPDH expression in ventricular tissue lysates from saline-sedentary, saline-exercise, doxorubicin-sedentary, and doxorubicin-exercise rat are shown (**A**). The bar graphs show protein levels of NLRP3 (**B**), cleaved-caspase-1 (**C**), and cleaved-IL-1β (**D**). Proteins were determined by immunoblotting and were normalized to Glyceraldehyde 3-phosphate dehydrogenase (GAPDH) for analysis. Data are presented as the fold change over the saline-sedentary group. The data are presented as mean ± SEM for six–eight rats per group. * *p*-value < 0.05, ** *p*-value < 0.01, *** *p*-value < 0.001 using one-way analysis of variance (ANOVA) and Turkey’s test. ns, not significant.

**Figure 6 ijms-25-09692-f006:**
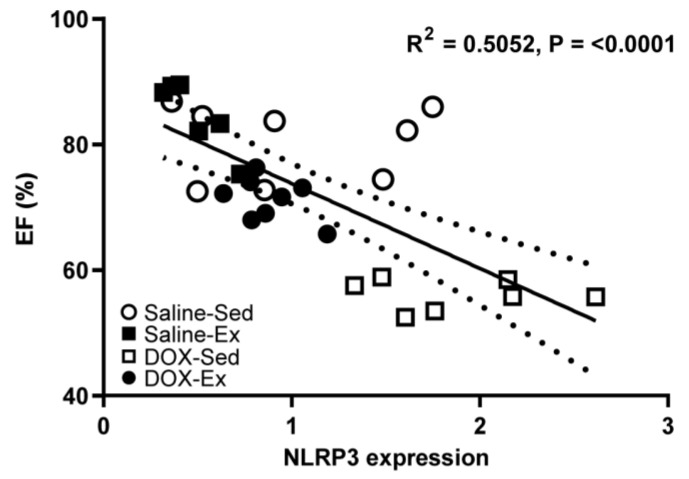
Relationship between percentage of ejection fraction (%EF) and the cardiac expression levels of NLRP3 from all experimental group combinations. Linear regression analysis indicated significant relations were found between NLRP3 expression and %EF (R^2^ = 0.5052). R^2^ represented the coefficient of determination. R represented Pearson’s correlation coefficient, with R values of 0–0.3, 0.3–0.5 and  >0.5 indicating weak, moderate, and strong correlation, respectively.

**Table 1 ijms-25-09692-t001:** Physical characteristics of saline-sedentary, saline-exercise, doxorubicin-sedentary, and doxorubicin-exercise rats.

Parameters	Saline Injection						Doxorubicin Injection					
	Sedentary (n = 8)			Exercise (n = 8)			Sedentary (n = 8)			Exercise (n = 8)		
Body weight, BW (g)	667	±	57.0	602	±	10.5 *	613	±	39.3 *	568	±	27.3 *
Heart weight, HW (g)	1.479	±	0.143	1.593	±	0.055 *	1.223	±	0.040 *	1.394	±	0.075 †
Tibial length, TL (cm)	4.54	±	0.05	4.53	±	0.03	4.50	±	0.04	4.53	±	0.04
Soleus weight, SW (g)	0.448	±	0.021	0.453	±	0.013	0.419	±	0.021	0.456	±	0.025
Kidney weight, KW (g)	1.969	±	0.044	1.864	±	0.055	1.774	±	0.035 *	1.659	±	0.052 †
HW/BW (×100)	0.221	±	0.010	0.265	±	0.004 *	0.219	±	0.005	0.246	±	0.007 ^#^
HW/TL (g/cm)	0.323	±	0.009	0.352	±	0.004 *	0.297	±	0.005 *	0.308	±	0.006 †
SW/BW (×100)	0.067	±	0.004	0.075	±	0.002	0.068	±	0.003	0.080	±	0.004
SW/TL (g/cm)	0.099	±	0.005	0.100	±	0.003	0.093	±	0.005	0.101	±	0.005
KW/BW (×100)	0.296	±	0.003	0.309	±	0.009	0.290	±	0.005	0.292	±	0.006
KW/TL (g/cm)	0.434	±	0.010	0.411	±	0.012	0.394	±	0.007 *	0.366	±	0.010 ^#^

Data are presented as mean ± SEM. *^,^†^,#^ *p*-value < 0.05, significantly different from the sedentary-saline, exercise-saline, and sedentary-doxorubicin group, respectively, using one-way analysis of variance (ANOVA) and Tukey’s test.

**Table 2 ijms-25-09692-t002:** Cardiac structural and functional changes by echocardiographic measurement.

Parameters	Saline Injection						Doxorubicin Injection					
	Sedentary (n = 8)			Exercise (n = 8)			Sedentary (n = 8)			Exercise (n = 8)		
LVPW; s (mm)	0.288	±	0.016	0.306	±	0.019	0.229	±	0.018	0.248	±	0.014 †
LVPW; d (mm)	0.306	±	0.025	0.274	±	0.014	0.239	±	0.016	0.245	±	0.009 †
IVS; s (mm)	0.308	±	0.010	0.338	±	0.012	0.231	±	0.011 *	0.248	±	0.015 †
IVS; d (mm)	0.185	±	0.013	0.221	±	0.008	0.187	±	0.012	0.199	±	0.008
LVID; s (mm)	0.331	±	0.017	0.303	±	0.020	0.450	±	0.031 *	0.394	±	0.015 †
LVID; d (mm)	0.594	±	0.020	0.586	±	0.027	0.659	±	0.049	0.606	±	0.019
LV mass (g)	1.135	±	0.114	1.121	±	0.084	1.034	±	0.072	0.984	±	0.048
LV mass/TL (g/cm)	0.250	±	0.025	0.249	±	0.017	0.227	±	0.015	0.218	±	0.011
Relative wall thickness	0.839	±	0.065	0.863	±	0.067	0.682	±	0.081	0.740	±	0.040

Data are presented as mean ± SEM. *^,^† *p*-value < 0.05, significantly different from the sedentary-saline and exercise-saline group, respectively, using one-way analysis of variance (ANOVA) and Tukey’s test. Left ventricular posterior wall thickness at end-systole (LVPW;s) and end-diastole (LVPW;d). Interventricular septal thickness at end-systole (IVS;s) and end-diastole (IVS;d). Left ventricular internal diameter at end-systole (LVID;s) and end-diastole (LVID;d).

## Data Availability

All the data obtained during this study are included in the manuscript. Additional information can be provided by the authors upon reasonable request.

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
