# Peer review of "Aerobic Exercise Attenuates Doxorubicin-Induced Cardiomyopathy by Suppressing NLRP3 Inflammasome Activation in a Rat Model"

_ijms, 2024, doi:10.3390/ijms25179692_

Round 1

Reviewer 1 Report

Comments and Suggestions for Authors

Minor comments:

1. Fig. 5. The legends of 4 groups should be added to western blot image in the panel A. 

2. Fig. 6. Weak correlations do not add value to results (put in supplementary). Put in evidence only strong and significant correlation with EF (inverse). Check Fig.6A. 

3. Line 42-44. Mention also endothelial function among the important cardiovascular benefits of regular exercise (quote PMID: 35643340, PMID: 37190153). Another limitation of this study is the lack of measurement of endothelial dysfunction in response to DOX-induced damage. 

Comments on the Quality of English Language

Minor editing is required

Author Response

Comments 1: Fig. 5. The legends of 4 groups should be added to western blot image in the panel A.

Response 1: Thank you for your suggestion. We have now included detailed labels for each of the four groups in the legend of panel A of the Western Blot image (Figure 5A) to improve clarity and ensure accurate interpretation.

Comments 2: Fig. 6. Weak correlations do not add value to results (put in supplementary). Put in evidence only strong and significant correlation with EF (inverse). Check Fig.6A.

Response 2: Thank you for your valuable feedback. We have revised the figure and the manuscript accordingly. The weak correlation graphs have been moved to the supplementary materials. We have updated the manuscript to emphasize the strong and significant inverse correlation between NLRP3 expression and cardiac ejection fraction (R² = 0.5052, P < 0.0001), as shown in Figure 6. The revised text on page 7, line 181-187 now states: “We subsequently investigated whether changes in NLRP3 expression could serve as a marker for cardiomyopathy and examined its relationship with cardiac function (Figure 6). Our analysis revealed that NLRP3 expression was strong inversely correlated with cardiac ejection fraction (R² = 0.5052, P = <0.0001) (Figure 6) but weak inversely correlated with blood pressure (R² = 0.1417, P = 0.0442) (Supplementary data, Figure S1A). Conversely, NLRP3 expression showed a direct weak correlation with histopathological score (R² = 0.2342, P = 0.0105) (Supplementary data, Figure S1B).” These changes have also been reflected in the figure legend of Figure 6.

Comments 3: Line 42-44. Mention also endothelial function among the important cardiovascular benefits of regular exercise (quote PMID: 35643340, PMID: 37190153). Another limitation of this study is the lack of measurement of endothelial dysfunction in response to DOX-induced damage.

Response 3: Thank you for your insightful comment. We acknowledge the limitation of our study in not measuring endothelial function, a critical aspect of cardiovascular health, and a significant benefit of regular exercise. In response, we have revised the manuscript to discuss the endothelial function and have cited the recommended articles (PMID: 37190153 and PMID: 35643340) on page 8, lines 206-210, and page 9, lines 231-237, respectively.

Reviewer 2 Report

Comments and Suggestions for Authors

This animal study examined the hypothesis that exercise intervention can attenuate doxorubicin (DOX)-induced cardiotoxicity by modulating nucleotide-binding oligomerization domain-like receptor protein 3 (NLRP3) inflammasome activity. The authors demonstrated that regular exercise confers protection against DOX-induced cardiotoxicity by suppressing NLRP3 inflammasome activation in the heart, underscoring its anti-inflammatory role, although it did not entirely prevent DOX-induced hypotension. They have mentioned that further research should explore the temporal dynamics and interactions among exercise, pyroptosis, and other pathways in DOX-induced cardiotoxicity to enhance translational applications in cardiovascular medicine.

This study is clinically significant, and the reviewer considers that the authors have conducted the research effectively. The reviewer has the following comments: 

Major comment:

1.     The study outlines a preventive protocol. In a clinical context, the administration of DOX should be planned in advance to allow for the incorporation of exercise. If regular exercise is discontinued, does the suppression of NLRP3 inflammasome activation in the heart also cease? 

Minor comment:

2.     In Tables 1 and 2, the lines under "Body weight, BW (g)" and "LVPW; s (mm)" are unnecessary and should be removed, respectively. 

Author Response

Comments 1: The study outlines a preventive protocol. In a clinical context, the administration of DOX should be planned in advance to allow for the incorporation of exercise. If regular exercise is discontinued, does the suppression of NLRP3 inflammasome activation in the heart also cease?

Response 1: Thank you for your insightful comment. You have highlighted a crucial aspect of our study regarding the dependency of NLRP3 inflammasome suppression on continuous exercise. We appreciate this observation, as it addresses an important consideration in translating our findings into clinical practice.

Our study primarily focuses on the preventive role of exercise prior to DOX administration, emphasizing the need for early intervention to mitigate the potential irreversibility of cardiotoxic effects. However, we acknowledge that the benefits of exercise, particularly in suppressing NLRP3 inflammasome activation, may indeed be transient if regular physical activity is discontinued.

We have included a discussion on this aspect in the revised manuscript. Specifically, we recognize that while exercise before DOX treatment is effective in reducing cardiac inflammation and maintaining cardiovascular health, the persistence of these benefits depends on the continuity of the exercise regimen. If exercise is not maintained, the suppression of NLRP3 inflammasome activation and its associated protective effects may diminish over time.

To address this limitation, we emphasize the need for further research to investigate the long-term sustainability of exercise-induced benefits and the impact of intermittent or discontinued exercise on NLRP3 inflammasome activity. Understanding these dynamics is crucial for developing practical and effective exercise protocols in clinical settings, where consistent physical activity may be challenging for patients.

This addition is detailed on page 11, lines 345-356 in the revised manuscript. We believe this clarification enhances the clinical relevance of our findings and underscores the importance of sustained exercise interventions in managing DOX-induced cardiotoxicity.

Comments 2: In Tables 1 and 2, the lines under "Body weight, BW (g)" and "LVPW; s (mm)" are unnecessary and should be removed, respectively.

Response 2: Thank you for your suggestion. We have removed the bottom border lines in Tables 1 and 2, as you suggested.

Reviewer 3 Report

Comments and Suggestions for Authors

In the present study explored the interactions between exercise, doxorubicine-induced cardiotoxicity, and the NLRP3 inflammasome in a male rat model . They found that  regular exercise in preventing the upregulation of NLRP3 inflammasome activation during doxorubicine-induced cardiotoxicity

This is an original research. Overall the manuscript is well written. The experimental protocol well conducted and explained

I have only minor suggestions

Discussion: the discussion paragraph appears to be too bulky. The authors should focus on the interpretation of the results, avoiding repeating concepts already expressed in the introductory paragraph.

Author Response

Comments 1: In the present study explored the interactions between exercise, doxorubicine-induced cardiotoxicity, and the NLRP3 inflammasome in a male rat model . They found that  regular exercise in preventing the upregulation of NLRP3 inflammasome activation during doxorubicine-induced cardiotoxicity

This is an original research. Overall the manuscript is well written. The experimental protocol well conducted and explained

I have only minor suggestions

Discussion: the discussion paragraph appears to be too bulky. The authors should focus on the interpretation of the results, avoiding repeating concepts already expressed in the introductory paragraph.

Response 1: Thank you for your constructive feedback. We appreciate your suggestion to streamline the discussion section for clarity and conciseness. In response, we have revised the discussion by focusing more directly on the interpretation of our results and minimizing the repetition of concepts already covered in the introduction.

Specifically, we have condensed the discussion to emphasize the key findings of our study, particularly the relationship between exercise and the suppression of NLRP3 inflammasome activation in the context of DOX-induced cardiotoxicity. We have removed redundant background information and ensured that the discussion stays closely aligned with the novel insights our research provides.

The revised discussion section has been updated accordingly, and we believe this revision will enhance the overall readability and impact of the manuscript. Thank you once again for your valuable input.

Round 2

Reviewer 2 Report

Comments and Suggestions for Authors

This reviewer has no further comment.